# Pediatric Stroke due to Thoracic Outlet Syndrome Treated with Thrombolysis and Thrombectomy: A Case Report

**DOI:** 10.3390/children9060875

**Published:** 2022-06-12

**Authors:** Dhanalakshmi Angappan, McKinnon Garrett, Candice Henry, Art Riddle, Jenny L. Wilson

**Affiliations:** 1Department of Pediatric Neurology, Oregon Health & Science University, Portland, OR 97239, USA; angappan@ohsu.edu (D.A.); garremac@ohsu.edu (M.G.); riddlea@ohsu.edu (A.R.); 2Department of Radiology, Oregon Health & Science University, Portland, OR 97239, USA; henryca@ohsu.edu

**Keywords:** pediatric stroke, endovascular treatment, thrombectomy, thoracic outlet syndrome, ischemic stroke

## Abstract

Thoracic outlet syndrome (TOS) is a condition that results from the compression of neurovascular structures as they exit the thorax. Arterial ischemic stroke can occur in TOS due to retrograde embolism from the subclavian artery. We describe a 15-year-old girl who presented with left hemiplegia after 2 weeks of right arm numbness and tingling. Imaging showed an acute ischemic stroke due to a right middle cerebral artery occlusion. She was treated with intravenous tissue plasminogen activator at 1.3 h and mechanical thrombectomy at 2.4 h with successful recanalization. Review of her neck computed tomography angiogram suggested a right subclavian artery aneurysm, and upper-extremity imaging also demonstrated distal thrombosis and fusion of right first and second ribs, which was consistent with thoracic outlet syndrome. Three days later, she underwent a right subclavian artery aneurysm repair, right brachial and ulnar artery thrombectomy, and first rib resection. Three months later, she demonstrated good neurologic recovery. TOS is an uncommon cause of stroke in children, which may be heralded by upper-extremity symptoms. Interventionalists should be aware of the possibility of vascular anomalies in children; however, this finding does not exclude the possibility of acute stroke intervention.

## 1. Introduction

Pediatric stroke is rare, occurring in 2–5/100,000 children per year [1,2], but is a significant cause of mortality and disability, ranking among the top 10 causes of pediatric death and leaving the majority of survivors with permanent neurologic deficits [3,4]. The causes are markedly different from adult stroke and include cardiac disorders, vasculopathies, sickle cell disease, and infections [5].

Thoracic outlet syndrome (TOS) is a rare cause of stroke. In TOS, upper-extremity symptoms develop due to compression of the neurovascular bundle (subclavian artery, brachial plexus, subclavian vein) as it leaves the chest above the first rib and behind the clavicle [6]. There are three types of TOS: neurogenic, resulting from compression of the nerves and causing sensory changes and weakness of the arm (most common, 90–95%); venous, resulting from subclavian vein obstruction and causing arm swelling and cyanosis (2–5%); and arterial, which is due to emboli from the subclavian artery causing symptoms of arm ischemia (least common, 1–2%) [6]. Arterial TOS is frequently associated with a cervical rib, which occurs in 1% of the population and is rarely symptomatic. Acute ischemic stroke (AIS) has been reported in more than 30 individuals with arterial TOS [7] and is a rare cause of stroke in children [7,8,9,10,11,12,13]. We present a 15-year-old girl with AIS due to TOS, complicated by subclavian artery thrombosis and subsequent embolism, whose stroke was treated with intravenous thrombolysis and mechanical thrombectomy.

## 2. Case Report

A 15-year-old girl with a history of migraine, anxiety, depression, and attention deficit disorder woke up with a headache for which she took a dose of sumatriptan. Upon entering her classroom at 7:45 a.m., she collapsed and was unresponsive for about a minute. The paramedics arrived with initial exam findings of a left-sided facial droop, left upper- and lower-extremity weakness, a right gaze preference, and slurred speech. The patient also reported right-arm numbness and tingling for 2 weeks prior to presentation. Medication use included bupropion, lisdexamfetamine dimesylate, and sumatriptan, with the addition of venlafaxine two weeks prior. In the emergency department, her weight was 70 kg, height 175 cm, heart rate 78 beats per minute, and blood pressure 109/68 mmHg. Computed tomography (CT) of the head demonstrated a hyperdense right middle cerebral artery (MCA) without MCA territory hypodensity (Alberta Stroke Program Early CT Score of 10), and subsequent computed tomography angiogram (CTA) of the head and neck confirmed a distal right internal carotid artery and proximal middle cerebral artery occlusion (Figure 1). Her NIHSS (National Institute of Health Stroke Scale) at presentation was 19 (level of consciousness, gaze preference, hemianopia, dysarthria, facial droop, left-sided hemiparesis, neglect, and decreased sensation). The patient was given 0.9 mg/kg of intravenous tissue plasminogen activator (tPA) 1.3 h after symptom onset. She was within the window for endovascular treatment, so she was transferred directly to the angiography suite at our institution, where she underwent mechanical thrombectomy with a Trevo Stentriever, achieving thrombolysis in cerebral infarction reperfusion grade 3 (complete perfusion) after one pass, which was 2.4 h after symptom onset (Figure 1). 

During the cerebral angiogram, her right hand was noted to be cool with diminished radial pulse, and right arm systolic blood pressure was noted to be 40–50 points lower than in the left arm. Her post-thrombectomy NIHSS decreased to 5. 

Although not initially appreciated, further review of her neck CTA prompted by identification of her asymmetric upper-extremity hemodynamics suggested a right subclavian artery aneurysm. The next day, a right upper-extremity CTA demonstrated a partially thrombosed right subclavian artery aneurysm with a long segment occlusion distally including brachial and ulnar artery thrombosis as well as abnormally fused first and second ribs, consistent with arterial thoracic outlet syndrome (Figure 2). 

She was treated with heparin after her post-stroke CT of the head showed no hemorrhage. On hospital day 4, she underwent right first rib resection, repair of right subclavian artery aneurysm with great saphenous vein graft, and right axillary, brachial, and ulnar thrombectomy. Post-operative doppler ultrasound studies confirmed recanalization of the right upper-extremity vessels. 

Laboratory evaluation was notable for a normal coagulopathy panel, mild anemia (hemoglobin 11.4 g/dL, range 12–16), normal metabolic panel except a creatinine of 0.85 mg/dL (0.5–0.8), and three negative COVID PCRs. Her hemoglobin A1C and thyroid stimulating hormone were normal. Her urine drug screen was positive for amphetamines, consistent with her prescription medication. A transthoracic echocardiogram with bubble study was normal. Hypercoagulable evaluation was not performed, as an etiology for her stroke was established. Her hospital course was complicated by post-operative hypotension, which was treated with red blood cell transfusion. She was started on aspirin for secondary stroke prevention. At the time of discharge, her NIHSS had improved to 1. Upon follow up at 3 months, she had recovered except for mild left-foot dystonia (NIHSS 0, pediatric stroke outcome score 0.5, modified Rankin score 1).

## 3. Discussion

We present a 15-year-old girl with acute arterial ischemic stroke successfully treated with IV tPA and thrombectomy who was found to have TOS with subclavian artery aneurysm as the etiology. Stroke is an uncommon complication of arterial TOS. The proposed mechanism is subclavian artery compression leading to aneurysmal dilation and thrombus formation with retrograde propagation of thrombus or, more rarely, retrograde embolism [14]. In our patient, there was no evidence of retrograde clot propagation (i.e., the right vertebral artery origin is patent, see Figure 2A), so the presumed mechanism was retrograde embolism. When the right subclavian artery is affected either anterior or posterior circulation stroke may occur depending on whether the vertebral or carotid artery is involved. However, when the left subclavian artery is affected, the posterior circulation rather than anterior circulation is involved, as the left carotid artery arises from the aortic arch. 

Stroke due to arterial TOS typically occurs in young adults and is preceded by upper-extremity symptoms in about 81% [7]. Stroke caused by TOS in children is rare, with 10 children reported in the literature including the current patient (Table 1). 

Five cases demonstrated right MCA involvement, four involved the posterior circulation, and one case involved both territories. Similar to adults, most children (70%) described upper-extremity symptoms due to TOS prior to their stroke, suggesting that better recognition of the symptoms of TOS in children could prevent stroke in some cases. Critically, four (40%) of the children had a history of TIA or stroke prior to the stroke that resulted in the TOS diagnosis. If the initial stroke evaluation had detected TOS, recurrence may have been prevented. When evaluating TIA or embolic stroke of unknown source, the subclavian artery should be considered as a potential origin. In children with stroke, physicians should routinely ask about preceding upper-extremity symptoms, evaluate for decreased or absent upper-extremity pulses or discordant blood pressures between limbs, and look for rib anomalies on available plain films. If any of these findings are present, vascular imaging should include the subclavian arteries. 

Adults with TOS often have headaches as a part of their symptomatology, which can resolve after treatment of TOS [6,17,18]. In some cases, headaches were provoked by arm or neck movements [17,18]. The mechanism is not clear but could be due to cerebral venous congestion [17,18]. Headache or migraine was reported in 30% of pediatric TOS cases in the literature, including ours [8,10]. Our patient had a long-standing history of migraine; however, she had no change in migraine pattern after TOS treatment and had a maternal family history of migraine. Thus, it was unlikely to be directly related in this case. Headache with or preceding stroke is not specific for TOS: about 38% of children with stroke of all causes have headache as a part of their clinical presentation [19]. However, headache provoked by arm abduction may be more specific to TOS.

There are three principles in the surgical management of TOS: (1) Decompression—typically resection of the cervical or first rib; (2) arterial resection—removal of aneurysm or vessel anomaly that could serve as a source of thrombus; and (3) distal revascularization—vascular reconstruction and removal of intraarterial clot [20]. There is no consensus on the timing of surgery. Acute limb ischemia necessitates surgery without delay. In the absence of limb ischemia, surgery may be delayed for a period of time while the patient is treated with anticoagulation. The clot burden, presence of subclavian artery aneurysm, recurrent symptoms despite conservative management, and medical stability are relevant factors in the timing of TOS surgery. Our patient had an aneurysm with a large clot burden, ongoing ischemic limb symptoms without acute ischemia at rest, and was clinically stable, so the decision was made to intervene surgically within a few days.

This is the first reported child with stroke due to TOS treated with IV tPA followed by thrombectomy. While IV tPA is standard of care in adults with acute stroke presenting within 4.5 h [21], data for IV tPA use in children remain limited to observational studies. The 2008 AHA pediatric stroke guidelines recommended against IV tPA use in children with the possible exception of adolescents who otherwise meet adult criteria [22]. Nonetheless, IV tPA does not appear to be less safe in children than in adults [23], and most tertiary pediatric hospitals in the United States have protocols for use of IV tPA in children with acute stroke [24,25].

Given strong evidence that endovascular therapies such as thrombectomy significantly benefit select adults with large vessel occlusion [26], there is increasing interest in the use of these therapies for children with stroke. Although there are no randomized controlled data, mechanical thrombectomy may be safe in children [27], and guidelines suggest that thrombectomy may be appropriate in some children with large vessel occlusion who otherwise have a poor prognosis [28,29]. However, children with stroke are at higher risk of vasculopathies, which may not be appropriate for thrombectomy or increase the risks of thrombectomy [30]. For example, the appearance of an inflammatory vasculopathy such as focal cerebral arteriopathy may mimic a large vessel occlusion; or, an interventionalist may encounter variant vessel anatomy in a child with congenital heart disease or a previously ligated carotid artery from extracorporeal membrane oxygenation treatment. In our patient, the subclavian abnormalities were distal to the path of the interventionalist’s catheter. Her treatment was successful and with no complications, a small final infarct size compared to the territory at risk, and good clinical recovery. However, subclavian vascular abnormalities may, in some cases, be near the origin of the carotid or vertebral arteries, such as in the case reported by Kataria et al. [9], potentially impacting thrombectomy. Our case highlights that pediatric stroke can be due to diverse and unexpected etiologies that may not be apparent at first presentation but may impact the risk of acute therapies. 

## 4. Conclusions

In children with embolic infarcts, the possibility of TOS causing retrograde embolization should be considered and may be heralded by upper-extremity symptoms. Recognition of this entity may prevent stroke recurrence. Acute therapies such as thrombolysis and thrombectomy may be appropriate in children with large vessel occlusion although the risk–benefit evaluation may be impacted by etiology.

## Figures and Tables

**Figure 1 children-09-00875-f001:**
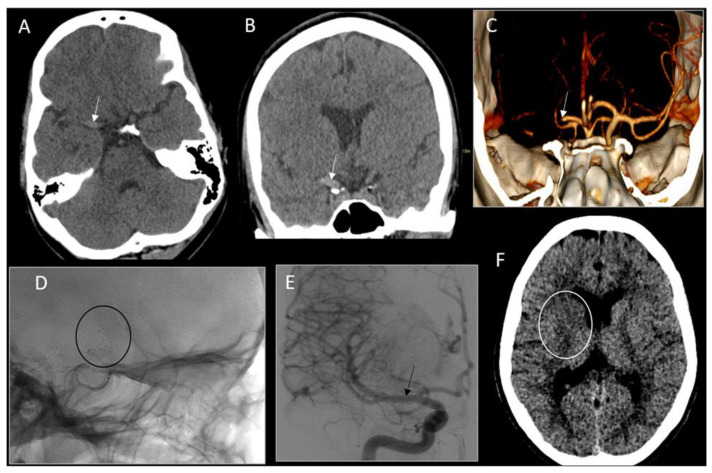
Neuroimaging of acute ischemic stroke and treatment. (**A**,**B**) Computed tomography angiogram of the head showing hyperdense middle cerebral artery (white arrows; (**A**) axial, (**B**) coronal). (**C**) Reformatted computed tomography angiogram of the head and 3D coronal demonstrating an abrupt cutoff of the right middle cerebral artery (white arrow). (**D**,**E**) Conventional cerebral angiogram showing stent retriever deployed across the right middle cerebral artery clot ((**D**) sagittal image, black circle) and, after successful clot retrieval, showing recanalized middle cerebral artery ((**E**) black arrow). (**F**) Non-contrast axial computed tomography of the head 24 h after intravenous tissue plasminogen-activator and thrombectomy showing ischemia limited to the right basal ganglia (loss of right caudate gray-white matter differentiation, white circle) without hemorrhagic conversion.

**Figure 2 children-09-00875-f002:**
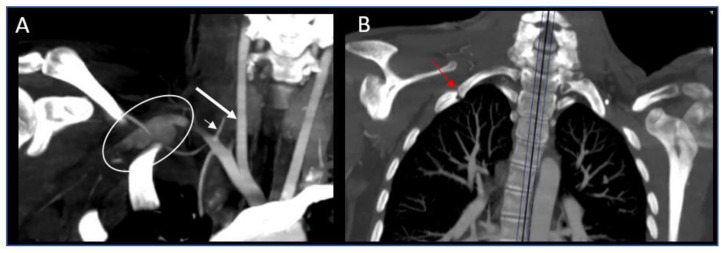
Computed tomography angiography of chest. (**A**) Right subclavian aneurysm (white circle) with lack of downstream intraluminal contrast compatible with thrombosis, presumed to have embolized to the right carotid artery (big arrow). Vertebral artery is patent (small arrow), coronal view. (**B**) Maximum intensity projection in bone windows highlighting abnormally fused right first and second ribs (white arrow), coronal view.

**Table 1 children-09-00875-t001:** Reported cases of pediatric stroke due to thoracic outlet syndrome.

Study	Age/ Sex	Risk Factors	Preceding Symptoms	Stroke Distribution	Treatment	Outcome
Current study	15 y F	Partially fused R 1st and 2nd ribs	Migraines R arm sensory changes	R MCA	IV tPA, cerebral thrombectomy Aspirin Rib resection, subclavian bypass, UE thrombectomy	Mild L foot dystonia
Aghamiri et al., 2022 [13]	15 y F	Heterozygous for plasminogen activator inhibitor and MTHFR	L arm sensory changes Stroke	R MCA	Cerebral thrombectomy x2 Aspirin, then long-term anticoagulation R UE thrombectomy	Not reported
Kuril et al., 2021 [10]	14 y M	Violinist and B cervical ribs	Migraines R arm sensory and color changes	B posterior circulation	Anticoagulation and aspirin Fasciotomy, UE thrombectomy Rib resection, subclavian aneurysm repair and bypass	Not reported
Strzelecka et al., 2018 [12]	8 y F	R cervical rib and R bifid 1strib	None	Posterior circulation	Anticoagulation Postponed rib excision until end of skeletal growth	R-sided weakness, intention tremor
Bains et al., 2014 [8]	12 y M	B hypoplastic 1st vs. cervical ribs	Chest pain Headache L hand cyanosis TIA	Posterior circulation	Rib excision	Mild dyscoordination, arm shakiness
Meumann et al., 2013 [7]	16 y F	B cervical ribs	R hand cyanosis R hand sensory changes TIA	R MCA	Anticoagulation Rib excision	Not reported
Kataria et al., 2012 [9]	14 y F	R cervical rib	Prior stroke	R MCA (old), posterior circulation (acute)	Anticoagulant and aspirin Rib excision	Minimal L-sided weakness not impacting function
Sharma et al., 2010 [15]	18 y M	R cervical rib	none	R MCA	Rib excision	Not reported
Lee et al., 2007 [11]	15 y F	B cervical ribs	R arm sensory changes	R MCA	Anticoagulation UE thrombectomy and bypass, rib excision	L arm weakness
Blank et al., 1974 [16]	18 y F	B cervical ribs	R hand pain	Posterior circulation	None reported	Resolution of neurologic symptoms

Reported cases of pediatric arterial ischemic stroke due to thoracic outlet syndrome including age and sex, symptoms preceding the acute presentation, stroke distribution, treatment (acute stroke, secondary prevention and treatment of the thoracic outlet syndrome), and patient outcome. B, bilateral; F, female; IV tPA, intravenous tissue plasminogen activator; L, left; M, male; MCA, middle cerebral artery; R, right; TIA, transient ischemic attack; TOS, thoracic outlet syndrome; UE, upper extremity; Y, years.

## Data Availability

Not applicable.

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
