# Peer review of "Pediatric Stroke due to Thoracic Outlet Syndrome Treated with Thrombolysis and Thrombectomy: A Case Report"

_children, 2022, doi:10.3390/children9060875_

Round 1
Reviewer 1 Report
This is a case report of an adolescent with an MCA occlusion due to thoracic outlet syndrome. It is well written and the provided pictures, along with the legends, further strengthen the presentation. I have only minor comments (which could be added during proofs):
Please provide the status of the right vertebral artery. If it is occluded then the most possible mechanism is retrograde propagation of thrombus. If not then a (much rarer) retrograde embolism should be hypothesized (according to picture 2A this is however the most possible clinical scenario).
The surgical repair was performed on day 4 post mechanical thrombectomy in order to prevent a possible recurrence. In general and based on your experience, how soon after the diagnosis of a thrombosed aneurysm due to TOS would you suggest such an intervention?
Reviewer 2 Report
The authors reported an interesting case of a pediatric patient with acute ischemic stroke caused by thoracic outlet syndrome.
They reported that this was the first report of a pediatric patient with acute ischemic stroke caused by thoracic outlet syndrome treated by intravenous thrombolysis and intra-arterial thrombectomy.
As they mentioned, stroke is an uncommon complication of arterial thoratic outlet syndrome, which is rare in children.
I think that this study is very well-written, however, I have a question.
What are the patient's height and weight? A 15-year old children may have similar height and weight as an adult, and I think that it may make this study better.
Thank you for your hard work, and congratulations.
